# Semantic Segmentation of Urban Buildings from VHR Remote Sensing Imagery Using a Deep Convolutional Neural Network

**Yaning Yi** [1,2,†], **Zhijie Zhang** [3,†], **Wanchang Zhang** [1,*], **Chuanrong Zhang** [3], **Weidong Li** [3] **and Tian Zhao** [4]

1   Key Laboratory of Digital Earth Science, Institute of Remote Sensing and Digital Earth,
    Chinese Academy of Sciences, Beijing 100094, China
2   University of Chinese Academy of Sciences, Beijing 100049, China
3   Department of Geography, University of Connecticut, Storrs, CT 06269, USA
4   Department of Computer Science, University of Wisconsin, Milwaukee, WI 53211, USA
*   Correspondence: zhangwc@radi.ac.cn; Tel.: +86-10-8217-8131
†   The first two authors are contributed equally to the work presented and are considered as equal first authors
    of this manuscript.

**Abstract:** Urban building segmentation is a prevalent research domain for very high resolution (VHR) remote sensing; however, various appearances and complicated background of VHR remote sensing imagery make accurate semantic segmentation of urban buildings a challenge in relevant applications. Following the basic architecture of U-Net, an end-to-end deep convolutional neural network (denoted as DeepResUnet) was proposed, which can effectively perform urban building segmentation at pixel scale from VHR imagery and generate accurate segmentation results. The method contains two sub-networks: One is a cascade down-sampling network for extracting feature maps of buildings from the VHR image, and the other is an up-sampling network for reconstructing those extracted feature maps back to the same size of the input VHR image. The deep residual learning approach was adopted to facilitate training in order to alleviate the degradation problem that often occurred in the model training process. The proposed DeepResUnet was tested with aerial images with a spatial resolution of 0.075 m and was compared in performance under the exact same conditions with six other state-of-the-art networks—FCN-8s, SegNet, DeconvNet, U-Net, ResUNet and DeepUNet. Results of extensive experiments indicated that the proposed DeepResUnet outperformed the other six existing networks in semantic segmentation of urban buildings in terms of visual and quantitative evaluation, especially in labeling irregular-shape and small-size buildings with higher accuracy and entirety. Compared with the U-Net, the F1 score, Kappa coefficient and overall accuracy of DeepResUnet were improved by 3.52%, 4.67% and 1.72%, respectively. Moreover, the proposed DeepResUnet required much fewer parameters than the U-Net, highlighting its significant improvement among U-Net applications. Nevertheless, the inference time of DeepResUnet is slightly longer than that of the U-Net, which is subject to further improvement.

**Keywords:** semantic segmentation; urban building extraction; deep convolutional neural network; VHR remote sensing imagery; U-Net

---

## 1. Introduction

One of the fundamental tasks in remote sensing is building extraction from remote sensing imagery. It plays a key role in applications such as urban construction and planning, natural disaster and crisis management [1–3]. In recent years, owing to the rapid development of sensor technology,

very high resolution (VHR) images with spatial resolution from 5 to 30 cm have become available [4], making small-scale objects (e.g., cars, buildings and roads) distinguishable and identifiable via semantic segmentation methods. Semantic segmentation as an effective technique aims to assign each pixel in the target image into a given category [5]; therefore, it was quickly developed and extensively applied to urban planning and relevant studies including building/road detection [6–8], land use/cover mapping [9–12], and forest management [13,14] with the emergence of a large number of publicly available VHR images.

In previous research, some machine learning methods were adopted to enhance the performance of VHR semantic segmentation with focus on the feature learning methods [15–18]. Song and Civco [19] adopted the support vector machine (SVM) with the shape index as a feature to detect roads in urban areas. Tian et al. [20] applied the random forest classifier to classify wetland land covers from multi-sensor data. Wang et al. [21] used the SVM-based joint bilateral filter to classify hyperspectral images. Das et al. [22] presented a probabilistic SVM to detect roads from VHR multispectral images with the aid of two salient features of roads and the design of a leveled structure. As pointed by Ball et al. [15], traditional feature learning approaches can work quite well, but several issues remain in the applications of these techniques and constrain their wide applicability.

The last few years witnessed the progress of deep learning, which has become one of the most cutting-edge and trending technologies thanks to hardware development of graphics processing unit (GPU). Owing to the successful application of deep convolutional neural network (DCNN) in object detection [23–25], image classification [26,27] and semantic segmentation [28–31], deep learning was introduced to remote sensing field for resolving the classic problems in a new and efficient way [32]. DCNN was adopted in many traditional remote sensing tasks, such as data fusion [33], vehicle detection [34,35] and hyperspectral classification [36,37]. As for building extraction, many DCNN-based methods have been proposed by many researchers [38–40]. For example, Saito et al. [41] directly extracted roads and buildings from raw VHR remote sensing image by applying a single convolutional neural network, and an efficient method to train the network for detecting multiple types of objects simultaneously was proposed. Marmanis et al. [4] proposed a trainable DCNN for image classification by combining semantic segmentation and edge detection, which significantly improved the classification accuracy. Bittner et al. [42] proposed the Fused-FCN4s model consisting of three parallel FCN4s networks to learn the spatial and spectral building features from three-band (red, green, blue), panchromatic and normalized digital surface model (nDSM) images. Vakalopoulou et al. [43] combined the SVM classifier and the Markov random field (MRF) model as a deep framework for building segmentation with Red-Green-Blue and near-infrared multi spectral images in high resolution. In contrast to feature learning approaches, deep learning approaches took advantage of several significant characteristics as summarized in Ball et al. [15]. However, adopting very successful deep networks to fit remote sensing imagery analysis can be challenging [15].

Very recent studies indicated that a deeper network would have a better performance when it came to object detection, visual recognition and semantic segmentation tasks. However, the deeper the network, the more significant the issues such as vanishing gradients. In order to account for this, He et al. [44] presented a deep residual learning approach, which reformulated the layers as learning residual functions with reference to the layer inputs, instead of learning unreferenced functions and achieved training of residual nets of 152 layers. This is eight times deeper than the VGG network while still maintaining lower complexity. Ronneberger et al. [30] presented a network and training strategy named U-Net, which performed data augmentation to make efficient use of annotated samples. High-level semantic information and low-level detailed information were combined by using the concatenate operation, and such a network can be trained in an end-to-end fashion from very few training images and still outperform the previous best approach.

In this paper, an end-to-end deep convolutional neural network (denoted as DeepResUnet) was proposed to complement semantic segmentation at pixel scale on urban buildings from VHR remote sensing imagery. Since according to the literature [15,26], a deeper network would have better

performance for semantic segmentation, we decided to follow network structure that enables the existence of larger number of layers in the network without running into training problems, thus the idea of residual learning is adopted in our network. Following the basic structure of U-Net, the proposed DeepResUnet contains two sub-networks: a cascade down-sampling network which extracts feature maps of buildings from the VHR image; and an up-sampling network which reconstructs the extracted feature maps of buildings back to the same size of the input VHR image. The deep residual learning approach was adopted to facilitate training in order to alleviate the degradation problem that often occurred in the model training process, and finally a softmax classifier was added at the end of the proposed network to obtain the final segmentation results.

To summarize, the main contributions of this paper are as follows. First, an end-to-end deep convolutional neural network, i.e., DeepResUnet, was proposed for complex urban building segmentation at pixel scale with three-band (red, green, blue) VHR remote sensing imagery. No additional data or any post-processing methods were adopted in this study. Second, in the DeepResUnet, the residual block (ResBlock) was designed as the basic processing unit to optimize model training and deep residual learning approach was applied to alleviate gradient-related issues. Third, in addition to comparing the performance of different deep models, the applicability of deep models was also explored by testing the trained models in a new urban area. Results indicated that DeepResUnet has the ability to identify urban buildings and it can be applied to dense urban areas such as big cities and even megacities. The purpose of this paper is not just to come up with a novel approach with better performance, it means more than just a higher accuracy. Our work can generate raw data (e.g., building boundaries) for geographical analysis such as urban planning and urban geography study. Only with more accurate raw data to begin with can those geographical analysis be accurate and instructive. We also used another totally new dataset to test and show that our proposed method is transferable with other datasets and can still maintain high performance, which means this method can have a very wide range of application. Last but not least, we hope that our proposed network structure can inspire scholars to build an even greater network.

Following the introduction, the remainder of this paper is arranged as follows: Section 2 introduces the architecture of the proposed DeepResUnet, with a focus on ResBlock, the down-sampling network, and up-sampling network. Detailed implementations of the DeepResUnet, extensive experimental results, and comparisons with other existing networks are presented in Section 3. The discussion is provided in Section 4, followed by Section 5 with the conclusions.

## 2. Methodology

DeepResUnet is an end-to-end DCNN that follows the basic structure of U-Net. DeepResUnet contains two sub-networks: A cascade down-sampling network for extracting building feature maps from the VHR image, and an up-sampling network for reconstructing the extracted building feature maps back to the same size of input VHR image. To reduce gradient degradation in model training, the deep-residual-learning approach was adopted in model training and a softmax classifier was used at the end of the network to obtain the final segmentation results. Figure 1 shows the detailed architecture of the proposed DeepResUnet network.

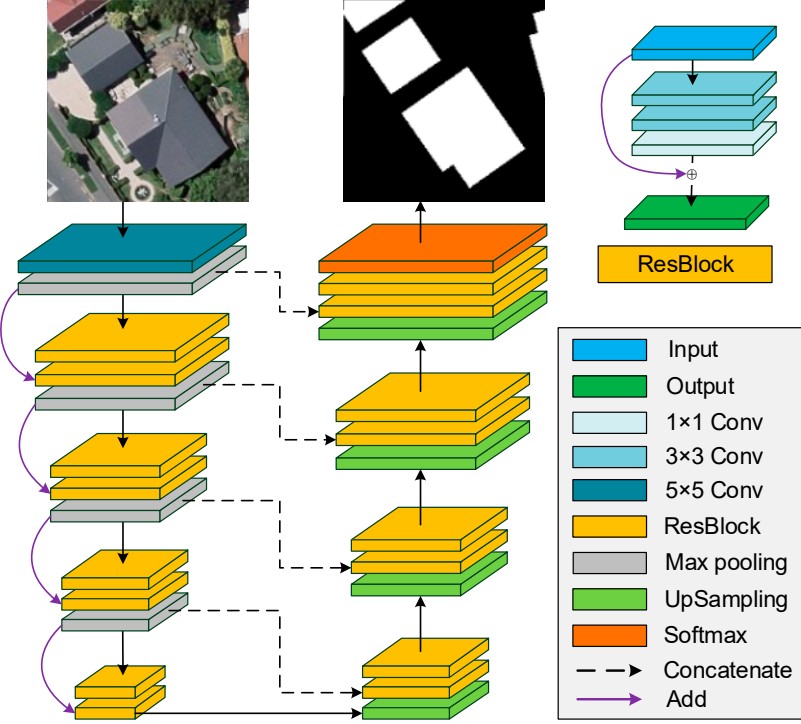

**Figure 1.** The architecture of the proposed DeepResUnet network. The input layer is an aerial image with three channels (red, green, blue) and the output is a binary segmentation map. The left part represents a down-sampling sub-network and the right part represents an up-sampling sub-network.

Table 1 presents the detailed information of DeepResUnet network. The input layer is a VHR aerial image with three channels (red, green, blue), and the output image is a binary segmentation map in which the pixel in white denotes the building and the pixel in black denotes the background. Similar to the U-Net, the architecture of DeepResUnet is mostly symmetrical but much deeper than that of U-Net. To accelerate the training, the batch normalization [45] layer was used after each convolutional layer. Note that no fully connected layers were used in DeepResUnet.

In the following subsections, we first give a brief description of the ResBlock architecture used in DeepResUnet and then provide the down-sampling and up-sampling sub-networks in detail.

**Table 1.** The architecture of the proposed DeepResUnet network.

| Name | Kernel Size | Stride | Pad | Output Size |
|------|------------|--------|-----|-------------|
| Down-sampling network | | | | |
| Input | – | – | – | $256 \times 256 \times 3$ |
| Conv_1 | $5 \times 5$ | 1 | 2 | $256 \times 256 \times 128$ |
| Pooling_1 | $2 \times 2$ | 2 | 0 | $128 \times 128 \times 128$ |
| ResBlock_1 | $3 \times 3/3 \times 3/1 \times 1$ | 1 | 1 | $128 \times 128 \times 128$ |
| ResBlock_2 | $3 \times 3/3 \times 3/1 \times 1$ | 1 | 1 | $128 \times 128 \times 128$ |
| Add_1 | – | – | – | $128 \times 128 \times 128$ |
| Pooling_2 | $2 \times 2$ | 2 | 0 | $64 \times 64 \times 128$ |
| ResBlock_3 | $3 \times 3/3 \times 3/1 \times 1$ | 1 | 1 | $64 \times 64 \times 128$ |
| ResBlock_4 | $3 \times 3/3 \times 3/1 \times 1$ | 1 | 1 | $64 \times 64 \times 128$ |
| Add_2 | – | – | – | $64 \times 64 \times 128$ |
| Pooling_3 | $2 \times 2$ | 2 | 0 | $32 \times 32 \times 128$ |
| ResBlock_5 | $3 \times 3/3 \times 3/1 \times 1$ | 1 | 1 | $32 \times 32 \times 128$ |
| ResBlock_6 | $3 \times 3/3 \times 3/1 \times 1$ | 1 | 1 | $32 \times 32 \times 128$ |
| Add_3 | – | – | – | $32 \times 32 \times 128$ |
| Pooling_4 | $2 \times 2$ | 2 | 0 | $16 \times 16 \times 128$ |
| ResBlock_7 | $3 \times 3/3 \times 3/1 \times 1$ | 1 | 1 | $16 \times 16 \times 128$ |
| ResBlock_8 | $3 \times 3/3 \times 3/1 \times 1$ | 1 | 1 | $16 \times 16 \times 128$ |
| Add_4 | – | – | – | $16 \times 16 \times 128$ |

**Table 1.** *Cont.*

| Name | Kernel Size | Stride | Pad | Output Size |
|------|-------------|--------|-----|-------------|
| Up-sampling network | | | | |
| UpSampling_1 | $2 \times 2$ | 2 | 0 | $32 \times 32 \times 128$ |
| Concat_1 | – | – | – | $32 \times 32 \times 256$ |
| Conv_1U | $1 \times 1$ | 1 | 0 | $32 \times 32 \times 128$ |
| ResBlock_1U | $3 \times 3/3 \times 3/1 \times 1$ | 1 | 1 | $32 \times 32 \times 128$ |
| ResBlock_2U | $3 \times 3/3 \times 3/1 \times 1$ | 1 | 1 | $32 \times 32 \times 128$ |
| UpSampling_2 | $2 \times 2$ | 2 | 0 | $64 \times 64 \times 128$ |
| Concat_2 | – | – | – | $64 \times 64 \times 256$ |
| Conv_2U | $1 \times 1$ | 1 | 0 | $64 \times 64 \times 128$ |
| ResBlock_3U | $3 \times 3/3 \times 3/1 \times 1$ | 1 | 1 | $64 \times 64 \times 128$ |
| ResBlock_4U | $3 \times 3/3 \times 3/1 \times 1$ | 1 | 1 | $64 \times 64 \times 128$ |
| UpSampling_3 | $2 \times 2$ | 2 | 0 | $128 \times 128 \times 128$ |
| Concat_3 | – | – | – | $128 \times 128 \times 256$ |
| Conv_3U | $1 \times 1$ | 1 | 0 | $128 \times 128 \times 128$ |
| ResBlock_5U | $3 \times 3/3 \times 3/1 \times 1$ | 1 | 1 | $128 \times 128 \times 128$ |
| ResBlock_6U | $3 \times 3/3 \times 3/1 \times 1$ | 1 | 1 | $128 \times 128 \times 128$ |
| UpSampling_4 | $2 \times 2$ | 2 | 0 | $256 \times 256 \times 128$ |
| Concat_4 | – | – | – | $256 \times 256 \times 256$ |
| Conv_4U | $1 \times 1$ | 1 | 0 | $256 \times 256 \times 128$ |
| ResBlock_7U | $3 \times 3/3 \times 3/1 \times 1$ | 1 | 1 | $256 \times 256 \times 128$ |
| ResBlock_8U | $3 \times 3/3 \times 3/1 \times 1$ | 1 | 1 | $256 \times 256 \times 128$ |
| Conv_5U | $1 \times 1$ | 1 | 0 | $256 \times 256 \times 2$ |
| Output | – | – | – | $256 \times 256 \times 2$ |

*2.1. ResBlock*

With increasing depth of deep neural networks, problems like vanishing gradients start to emerge. To resolve this issue, the deep residual framework (ResNet) was proposed to ensure the gradient be directly propagated from top to bottom of the network during the backward propagation [44]. Previous studies suggested that the residual framework can improve accuracy considerably with increased layer depth and is easier to optimize [46].

Formally, by denoting the input as $x_l$, the output of a residual unit as $x_{l+1}$ and the residual function as $F(\cdot)$, the residual unit can be expressed as:

$$x_{l+1} = F(x_l) + x_l \tag{1}$$

Inspired by the ResNet, the residual block (ResBlock) was designed as the basic processing unit in DeepResUnet to optimize model training. Figure 2 illustrates the structure of different types of Resblocks. In general, they are all shaped like a bottle neck. A $1 \times 1$ convolutional layer following two successive $3 \times 3$ convolutional layers in the ResBlock with ReLU as activation function between successive layers was designed to account for gradient degradation of image in the training process. For better performance, two successive $3 \times 3$ convolution kernels were adopted in DeepResUnet by following the suggestions from other studies [47,48]. It is worthwhile mentioning that the number of channels of the first $3 \times 3$ convolution layer was twice than that of the latter. A small number of channels of the first $3 \times 3$ convolutional layer can reduce model parameters without losing too much image information. Additionally, a $1 \times 1$ convolution layer was added to the ResBlock.

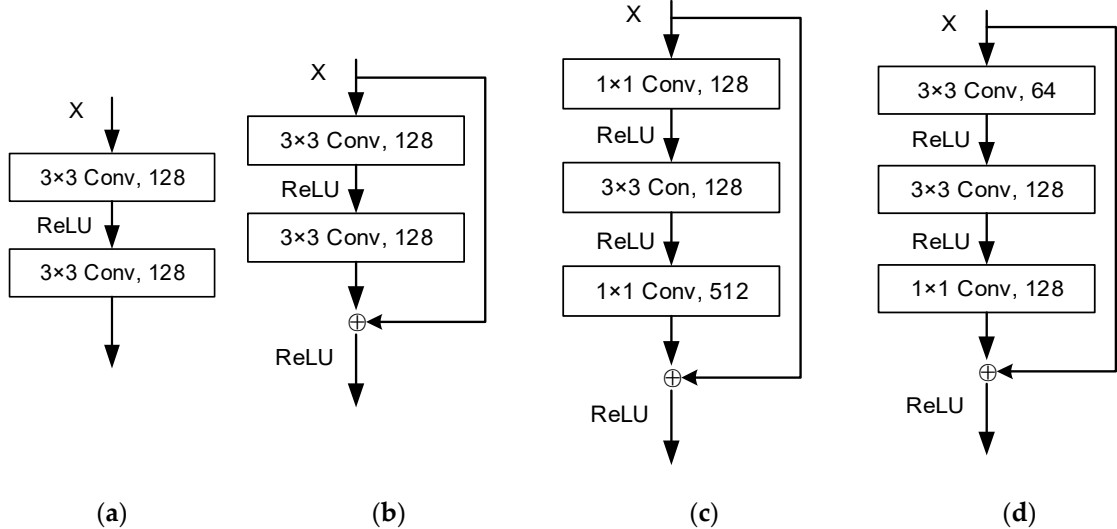

**Figure 2.** Illustration of the ResBlock structure. (**a**) plain neural unit used in U-Net; (**b**) basic residual unit used in ResNet-34; (**c**) "bottleneck" shaped residual unit used in ResNet-50/101/152; (**d**) the designed ResBlock used in DeepResUnet.

## 2.2. Down-Sampling Network

Inspired by ENet [49], early downscale-sampling method was employed after the input layer in the proposed DeepResUnet network. The assumption behind this is that the feature maps from the initial image layer contain adverse noise that would directly contribute to segmentation, which should be filtered. Therefore, max-pooling layer with the size of $2 \times 2$ was added to reduce the input size. It is worthwhile mentioning that although the pooling operation is capable of reducing the learning parameters while keeping scaling invariant, spatial information essential for pixelwise segmentation was indeed partially lost in this process [15]. To keep as much spatial information as possible, a $5 \times 5$ convolutional layer with 128 channels was added before the first max-pooling layer to gain a larger receptive field.

As exhibited in Figure 1, two successive ResBlock modules were set after the first max-pooling layer to obtain image features, and then a $2 \times 2$ max-pooling layer was added to reduce the learning parameters while keeping scaling invariant for enlarging the receptive field. To make better use of the previous features and propagate the gradient, the feature maps from the pooling layer were added to the output of two successive ResBlock modules in a residual manner. The detailed description of the down-sampling process is exhibited in Figure 3a. The input of the latter pooling layer was computed by:

$$y = f(f(P(x))) + P(x) \tag{2}$$

where $P(\cdot)$ represents the pooling function, $f(\cdot)$ represents the ResBlock operation, $x$ represents the input, and $y$ is the output which is used as the input for the subsequent pooling and up-sampling operations.

To effectively exploit image information, a down-sampling network with two successive ResBlock layers and one pooling layer repeated for three times was developed in DeepResUnet. Consequently, the size of the input image was reduced from $256 \times 256$ to $16 \times 16$. In addition, two successive ResBlock layers without pooling layer were employed at the end of down-sampling network to serve as a bridge connecting the down-sampling and up-sampling networks.

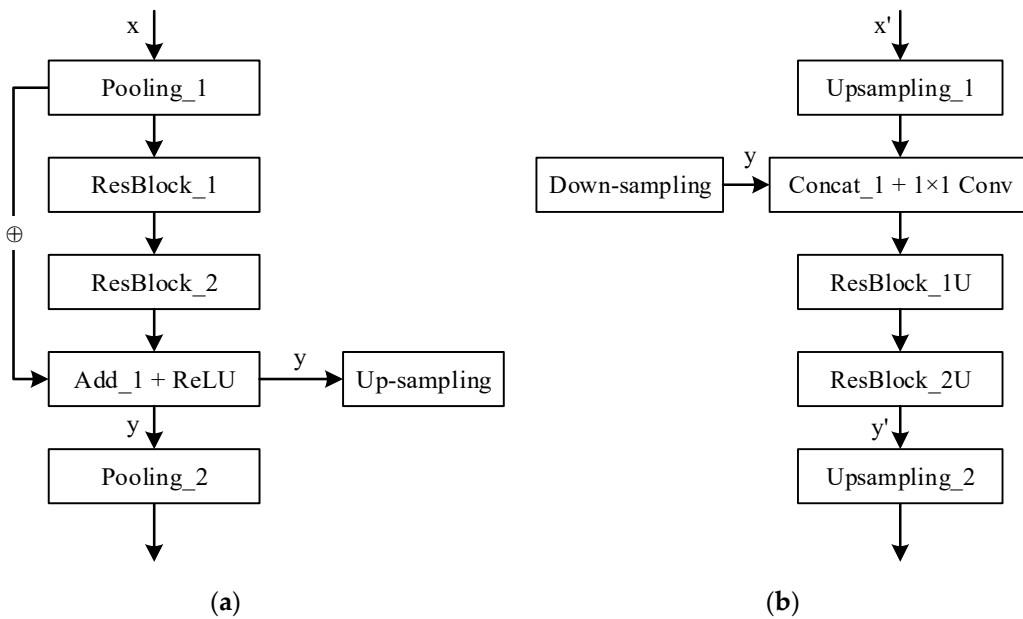

**Figure 3.** The details of the down-sampling (**a**) and up-sampling structures (**b**).

## 2.3. Up-sampling Network

Similar to the U-Net, our up-sampling network was symmetric to the down-sampling network. As illustrated in Figure 1, the up-sampling operation functions in recovering the details of feature maps. For pixelwise segmentation, aggregating multi-context information can effectively improve the performance of deep learning models [50]. The intuition is that low-level features contain finer details which can compensate for high-level semantic features [51]. In the up-sampling network, the low-level features at a pixel scale were propagated to the corresponding high levels in a concatenation manner, and then a $1 \times 1$ convolutional layer was employed to change the output channels. Subsequently, two successive ResBlock modules were added after the concatenation operation. As illustrated in Figure 3b, the output of ResBlock can be formulated as:

$$y' = f(f(f_{1\times1}(U(x') \otimes y)))$$  (3)

where $f_{1 \times 1}$ $(\cdot)$ is the $1 \times 1$ convolutional operation, $U(\cdot)$ is the up-sampling operation, $x'$ is the previous feature maps from the down-sampling network, and the symbol of $\otimes$ denotes the concatenation operation.

Different from the down-sampling network, the up-sampling network adopted the up-sampling layer instead of the max-pooling layer. Four up-sampling layers were used in the up-sampling network to facilitate reconstructing the feature maps to the same size as the input image. Following the last ResBlock layer in the up-sampling network, a softmax layer was used to derive the final segmentation maps.

## 3. Experiments and Results

In this section, we first describe the dataset used in the experiments and experimental setting. We then provide qualitative and quantitative comparisons of performances between DeepResUnet and other state-of-the-art approaches in semantic segmentation of urban buildings from the same data source (VHR remote sensing imagery).

### 3.1. Dataset

Large-scale training samples are required for deep learning models to learn various features. Aerial images with a spatial resolution of 0.075 m and the corresponding building outlines were

collected from a public source (https://data.linz.govt.nz) as our experiment dataset. The aerial images covered an urban area of Christchurch City, New Zealand, and were taken during the flying season (summer period) 2015 and 2016. Note that the aerial images had been converted into orthophotos (three-band, red-green-blue) by the provider and were provided in format of tiles. The pixel value of the aerial image varies from 0 to 255. The image we selected was mosaiced with 335 tiles covering 25,454 building objects, and the mosaic image is 38,656 × 19,463 pixels, as shown in Figure 4. To train and test the DeepResUnet, the mosaic image was split into two parts for training and testing, with almost equivalent areas including 12,665 and 12,789 building objects, respectively (Figure 4). Meanwhile, the building outlines corresponding to the aerial image, which were stored as polygon shapefiles, were similarly converted into a raster image and further sub-divided into two parts as was done to the aerial image. In the experiment, the building outlines were regarded as the ground truth to train and evaluate the methods. Note that here the elevation data like the digital surface model (DSM) were not used in the dataset, and data augmentation algorithms were not used in this study.

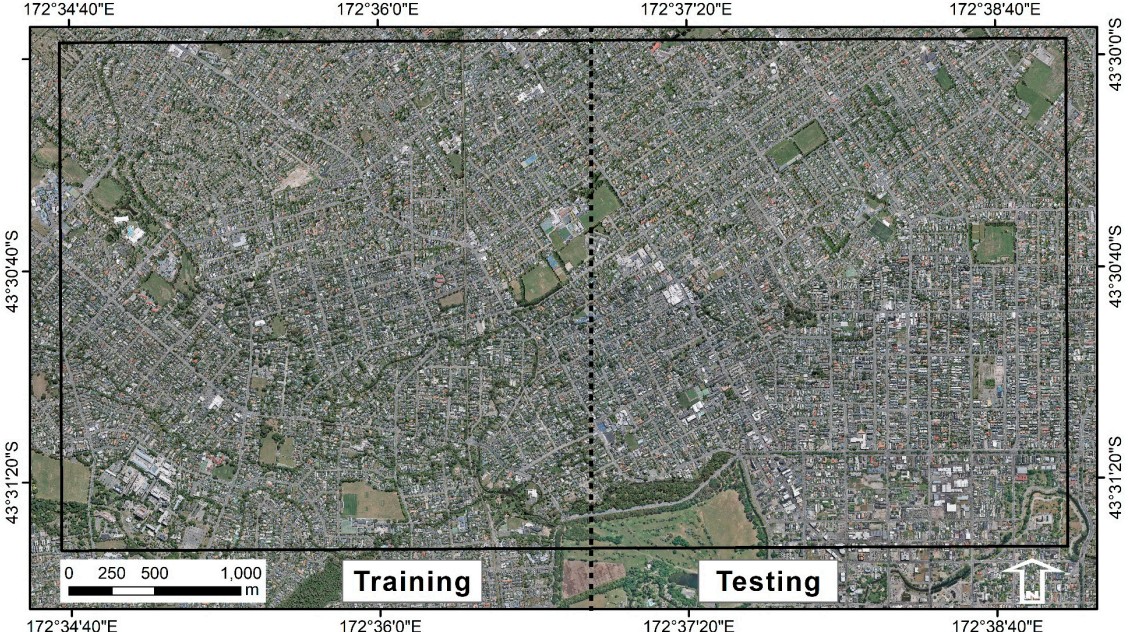

**Figure 4.** Overview of the dataset (an urban area of Christchurch City, New Zealand) used in the present study. The black dotted line separates the image into training and testing datasets for evaluating the performance of the proposed DeepResUnet by comparing it with other state-of-the-art deep learning approaches.

### 3.2. Experimental Setup

DeepResUnet was implemented in Keras using the Tensorflow framework as the backend. In the experiment, the aerial imagery of the training area was split into patches with the size of 256 × 256 by using a sliding window algorithm with a stride of 128. Thus, a total of 17,961 training patches of the same size (256 × 256 image block) were prepared. During training, 80% of training patches were used to train the models while the remaining 20% was used for cross-validation. The proposed network and other comparison ones were trained on a NVIDIA GeForce GTX 1080Ti GPU (11 GB RAM). The glorot normal initializer [52] was used to initialize weights and parameters of networks, and the cross-entropy loss was employed in training process. The Adam optimizer [53] was adopted to optimize the training loss. Due to the limited memory of GPU, the batch size of 6 was chosen in the experiment. The learning rate was set at 0.001 initially, but it was gradually reduced by a factor of 10 in every 11,970 iterations. In this experiment, the proposed DeepResUnet converged after 35,910 iterations.

During the inference phase, the aerial imagery of the test area was also split into patches with the size of 256 × 256 given the limitation of GPU memory. To reduce the impact of boundaries, a sliding window algorithm with a stride of 64 was applied to generate the overlap images and the predictions of overlap images were averaged as final segmentation results. For the purpose of clearly reflecting the performance of the DeepResUnet, no post-processing methods such as the filters [54] and conditional random field [55] were used in the study in order to guarantee that a fair comparison in terms of the pure network performance.

*3.3. Results*

To evaluate the performance of DeepResUnet, six existing state-of-the-art deep learning approaches (i.e., FCN-8s, SegNet, DeconvNet, U-Net, ResUNet and DeepUNet) were selected for comparison in the exact same experimental environment. Each network was trained from scratch, without using pretrained models, and all networks converged during training. The inference procedure of six existing deep learning approaches was the same as that of DeepResUnet. The overall results from different networks of a randomly selected test area are shown together in Figure 5.

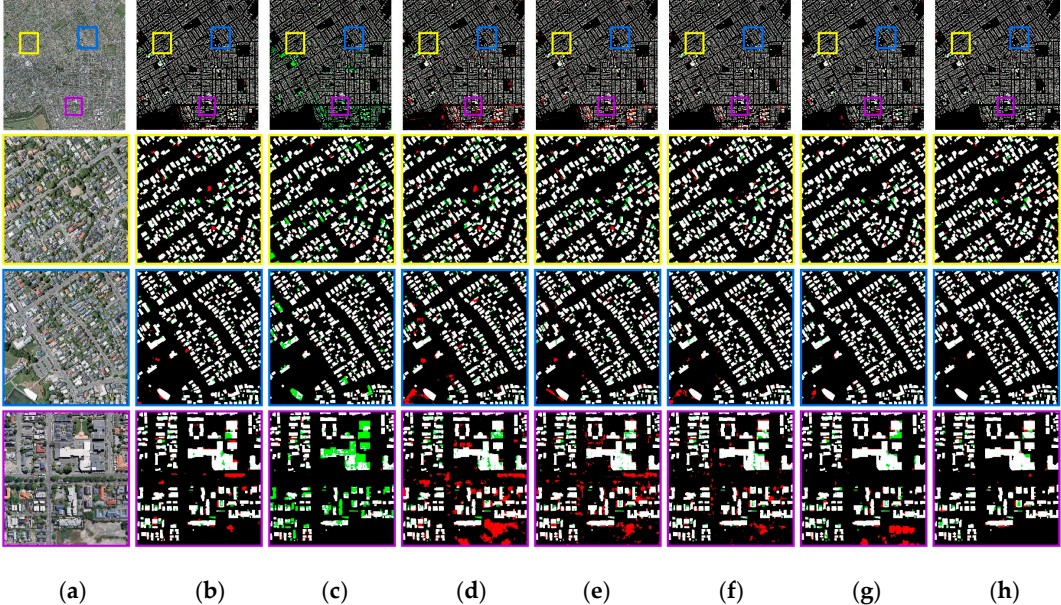

|   (**a**)   |   (**b**)   |   (**c**)   |   (**d**)   |   (**e**)   |   (**f**)   |   (**g**)   |   (**h**)   |

**Figure 5.** Visual comparison of segmentation results (an urban area of Christchurch City, New Zealand) obtained by seven approaches. The first row shows the overall results of a randomly picked test area generated by the seven networks, while the last three rows exhibit the zoomed-in results of the corresponding regions that were also randomly picked from the test area. In the colored images of the last three rows, the white, red and green colors represent true positive, false positive and false negative predictions, respectively. (**a**) Image. (**b**) FCN-8s. (**c**) SegNet. (**d**) DeconvNet. (**e**) U-Net. (**f**) ResUNet. (**g**) DeepUNet. (**h**) DeepResUnet.

By visual inspection, it appears that DeepResUnet outperformed the other approaches. As illustrated in Figure 5, the major parts of buildings were accurately extracted by DeepResUnet, while more false positives (red) and more false negatives (green) were found in the semantic segmentation of urban buildings by the other approaches, especially SegNet, DeconvNet and U-Net. Visually, FCN-8s, ResUNet and DeepUNet models performed better than other three approaches selected for comparison, but they still did not accurately identify small-size buildings in dense building area and FCN-8s model often misclassified roads as buildings. It is worthwhile mentioning that ResUNet and DeepUNet models as the improved versions of U-Net, outperformed the original U-Net model, similar to the proposed DeepResUnet, but more false positives (red) occurred in their segmentation results. There

were few misclassifications (false positive) in the segmentation results of SegNet model, but many false negatives (green) appeared in the segmentation results, indicating that many buildings were not accurately identified by SegNet. The DeconvNet and U-Net models frequently misclassified roads and bare surfaces as buildings, and many false positives (red) were obtained in the corresponding segmentation results, implying that these models did not make full use of the image features. Overall, DeepResUnet outperformed the other six approaches with less false negatives and false positives in the semantic segmentation image of urban buildings.

For further comparison, the partial results of the test area for semantic segmentation of urban buildings are presented in Figure 6. It is also clear that all the seven tested approaches can identify the regular shaped buildings in general with acceptable accuracy, such as rectangle and square shaped buildings. However, for small-size and irregularly shaped buildings (as shown in Figure 6), DeepResUnet had very competitive performance on better preservation of patch edges, followed by DeepUNet, while more false positives (red) and false negatives (green) were obtained by the other approaches. Among these approaches, SegNet, DeconvNet and U-Net generated considerably more incomplete and inaccurate labelings than FCN-8s and ResUNet did. Although the proposed DeepResUnet was not yet perfect in the semantic segmentation of urban buildings, relatively more accurate extraction of building boundaries and relatively coherent building object labeling with fewer false positive returns made it rank in a high position among all the seven compared state-of-the-art deep networks.

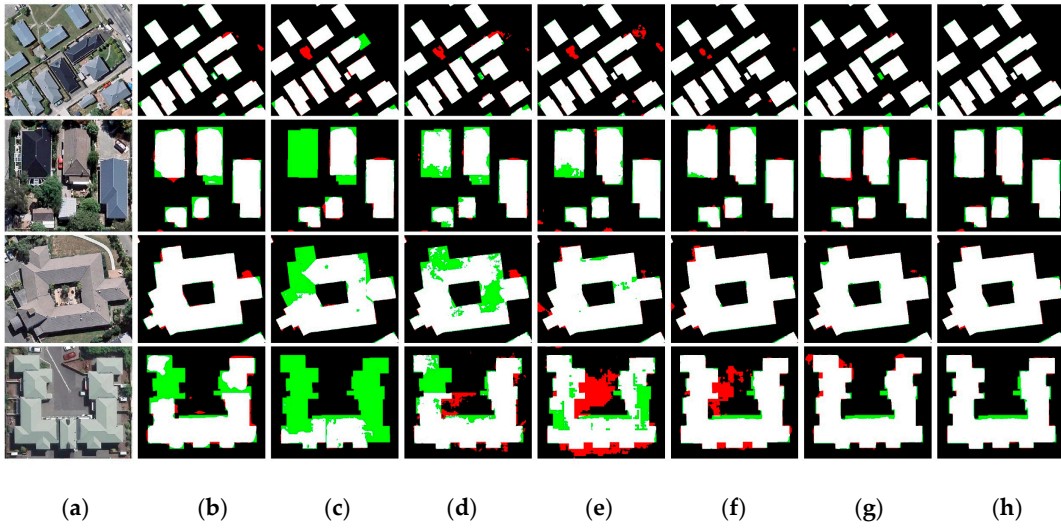

|     |     |     |     |     |     |     |     |
| --- | --- | --- | --- | --- | --- | --- | --- |
| (**a**) | (**b**) | (**c**) | (**d**) | (**e**) | (**f**) | (**g**) | (**h**) |

**Figure 6.** Visual comparison of segmentation results by seven approaches. In the colored images, the white, red and green colors represent true positive, false positive and false negative predictions, respectively. (**a**) Image. (**b**) FCN-8s. (**c**) SegNet. (**d**) DeconvNet. (**e**) U-Net. (**f**) ResUNet. (**g**) DeepUNet. (**h**) DeepResUnet.

To quantitatively evaluate the performance of DeepResUnet, five conventionally used criteria, including precision, recall, F1 score (F1), Kappa coefficient and overall accuracy (OA), were employed. The Kappa coefficient and OA are the global measures of segmentation accuracy [56]. Precision measures the percentage of matched building pixels in the segmentation map, while Recall represents the proportion of matched building pixels in the ground truth. F1 is the geometric mean between precision and recall, which is formulated as:

$$\text{F1} = 2 \times \frac{Precision \times Recall}{Precision + Recall} \tag{4}$$

As illustrated in Table 2, DeepResUnet had the best performance among all networks in terms of all the five criteria, followed by DeepUNet. All evaluation criteria were improved a considerable

amount in the test area by the proposed DeepResUnet in this study. Among these deep models, SegNet and DeconvNet had the worst performance, followed by U-Net, ResUNet and FCN-8s. It is worthwhile mentioning that both ResUNet and DeepUNet models had a better performance than the original U-Net model, indicating the effectiveness of the combined methods. Although most models achieved relatively high values in one of the evaluations metrics, none of them have good performance in all metrics. For instance, DeepResUnet outperformed SegNet in term of Precision index only by about 0.7%. However, in terms of Recall index, DeepResUnet outperformed SegNet more than by 12%. The proposed DeepResUnet had the best performance in terms of the Recall index compared with other six networks, indicating that DeepResUnet was more effective in suppressing false negatives in semantic segmentation of urban buildings. With respect to the F1 score, Kappa coefficient and OA, the proposed DeepResUnet in this study was still the best among all deep models. This further demonstrated the superiority of the proposed network in semantic segmentation of urban buildings from VHR remotely sensed imagery.

**Table 2.** Quantitative comparison of five conventionally used metrics obtained from the segmentation results (for an urban area of Christchurch City, New Zealand) by FCN-8s, SegNet, DeconvNet, U-Net, ResUNet, DeepUNet and the proposed DeepResUnet, where the values in bold format are the highest numbers for corresponding metrics.

| Models | Precision | Recall | F1 | Kappa | OA |
|---|---|---|---|---|---|
| FCN-8s [28] | 0.9163 | 0.9102 | 0.9132 | 0.8875 | 0.9602 |
| SegNet [29] | 0.9338 | 0.8098 | 0.8674 | 0.8314 | 0.9431 |
| DeconvNet [31] | 0.8529 | 0.9001 | 0.8758 | 0.8375 | 0.9413 |
| U-Net [30] | 0.8840 | 0.9190 | 0.9012 | 0.8709 | 0.9537 |
| ResUNet [51] | 0.9074 | 0.9315 | 0.9193 | 0.8948 | 0.9624 |
| DeepUNet [57] | 0.9269 | 0.9245 | 0.9257 | 0.9035 | 0.9659 |
| DeepResUnet | **0.9401** | **0.9328** | **0.9364** | **0.9176** | **0.9709** |

## 4. Discussion

### 4.1. About the DeepResUnet

DeepResUnet adopted the U-Net as its basic structure and meanwhile took advantages of deep residual learning by replacing the plain neural units of the U-Net with the residual learning units to facilitate the training process of the network. Although some combined approaches of the U-Net with the deep residual network have been reported in recent studies [51,57,58], significant differences can be found between their network architectures and that of ours.

First, in the network architecture, a new residual block, namely, Resblock, was designed as the basic processing unit to learn various representations of remote sensing images. ResBlock consisted of two successive $3 \times 3$ convolutional layers and a single $1 \times 1$ convolutional layer, which was designed to replace the basic residual unit that was used by other combined approaches of combining U-Net with the residual learning approach. Although a single $1 \times 1$ convolutional layer has been proved effective in dimension reduction [26], some important information for pixelwise segmentation might be compressed under the limited number of network layers and consequently would cause the loss of the transferred information, thus affecting the final performance of the deep network. For resolving this knotty problem, we made a tradeoff between the number of parameters and network layers by using two successive $3 \times 3$ convolutional layers and a single $1 \times 1$ convolutional layer in the ResBlock and the number of channels of the latter $3 \times 3$ convolution layer doubles that of the former. Hence, the structure of DeepResUnet is much deeper than that of U-Net.

Second, the main purpose of this study was to perform semantic segmentation of urban buildings from VHR remotely sensed imagery. To some extent, the proposed network was designed for pixel-level urban building semantic segmentation, and it may be applied to dense urban areas such as big cities and even megacities. For clearly and fairly reflecting the performance of different networks, any

post-processing operations, such as the filters or conditional random field, were not applied in the proposed DeepResUnet. Additionally, the infrared band data and DSM data, which are the two data sources that have much potential in improving the final performance if combined with aforementioned post-processing operations, were also not used in this study. In addition, the performance of these combined approaches (i.e., ResUNet [51] and DeepUNet [57]) was compared in the Section 3.3. Among all combined models, the proposed DeepResUnet model achieved the best performance, indicating the effectiveness of DeepResUnet. Note that those combined models obtained very high values compared with the original U-Net model; therefore, the combination of different methods will be a new trend to improve the performance in the future.

## 4.2. Effects of Resblock

To further confirm the effectiveness of Resblock, the performances of DeepResUnet (Baseline + Resblock) and its variants were compared and are summarized in Table 3. Here, baseline refers to the basic structure of our network which is "U" shaped. The basic residual unit refers to the structure used in ResNet-34. Bottleneck refers to the "bottleneck" shaped residual unit used in ResNet-50/101/152. Plain neural units are just convolution layers that do not use the residual structure. As shown in Table 3, the performance of Baseline + bottleneck was similar to that of Baseline + plain neural unit, close to that of Baseline + basic residual unit, indicating that the bottleneck cannot improve the performance of the baseline model. However, the Baseline + basic residual unit outperformed Baseline + plain neural unit, implying that the residual learning is effective in building semantic segmentation task. The poor performance of Baseline + bottleneck indicated that the $1 \times 1$ convolutional layer in the bottleneck may have a negative impact on information transmission, and more research is needed to confirm this. A comparison between Baseline + bottleneck and Baseline + basic residual unit, Baseline + Resblock (i.e., DeepResUnet) achieved better segmentation results. This implies the effectiveness of replacing the $1 \times 1$ convolutional layer in bottleneck with a $3 \times 3$ convolutional layer and also indicates that reducing the number of channels by half in the first $3 \times 3$ convolution layer has little to almost no effect on the accuracy of the segmentation result, but this procedure greatly reduces the number of parameters which would benefit the training process and create a more robust model.

**Table 3.** Comparisons of building segmentation results and model complexity among the different variants of DeepResUnet.

| Metrics | Baseline + Plain Neural Unit | Baseline + Basic Residual Unit | Baseline + Bottleneck | Baseline + Resblock (DeepResUnet) |
|---|---|---|---|---|
| Precision | 0.9234 | 0.9329 | 0.9277 | 0.9401 |
| Recall | 0.9334 | 0.9330 | 0.9321 | 0.9328 |
| F1 | 0.9283 | 0.9329 | 0.9299 | 0.9364 |
| Kappa | 0.9068 | 0.9129 | 0.9089 | 0.9176 |
| OA | 0.9669 | 0.9691 | 0.9677 | 0.9709 |
| Parameters (m) | 4.89 | 4.89 | 3.06 | 2.79 |
| Training time (second/epoch) | 1485 | 1487 | 1615 | 1516 |
| Inference time (ms/image) | 63.5 | 63.8 | 72.5 | 69.3 |

In terms of complexity, Baseline + Resblock requires fewer parameters than the other models because a small number of channels was applied in the first convolutional layer of Resblock (as exhibited in Figure 2). Although the structure of Resblock is similar to that of the bottleneck, Baseline + bottleneck requires a longer training time and inference time than Baseline + Resblock. In addition, Baseline + Resblock needs a slightly longer training time and inference time than Baseline + plain neural unit

and Baseline + basic residual unit, but generated better segmentation results. Overall, the structure of Resblock has an obvious effect in improving the performance of deep models.

### 4.3. Complexity Comparison of Deep Learning Models

In recent years, with the rapid advancement of the computer hardware, high-end GPU or GPU clusters have made network training easier, and some deeper networks have been proposed, such as DenseNets [59] and its extended network [60]. However, for experimental research and practical application, the trade-offs among the layer depth, number of channels, kernel sizes, and other attributes of the network must still be considered when designing the architectures of networks [61], given the concern of cost-effectiveness in training time and commercial cost.

To evaluate the complexity of DeepResUnet, the number of parameters, training time and inference time were compared with six existing state-of-the-art deep learning approaches (i.e., FCN-8s, SegNet, DeconvNet, U-Net, ResUNet and DeepUNet). It is worthwhile mentioning that the running time of deep models including training and testing time can be affected by many factors [62], such as the parameters and the model structure. Here, we simply compared the complexity of deep models. As shown in Table 4, DeconvNet has the largest number of parameters, and the longest training time and inference time among all models. The numbers of parameters of DeepResUnet are much fewer than most networks except the DeepUNet, because DeepUNet adopted a very small convolutional channel (each convolutional layer with 32 channels). Even though the proposed DeepResUnet followed the basic structure of U-Net, the number parameters of U-Net are nearly eleven times higher than that of DeepResUnet. However, DeepResUnet requires a longer training time and inference time than U-Net and its combined networks. The main reason may be that the deep residual learning may have a negative effect on model operations. The training time and inference time of ResUNet are longer than those of U-Net, also indicating the negative impact of deep residual learning. Additionally, the structure of DeepResUnet is much deeper than those of deep networks, which may also increase the operation time of DeepResUnet. Compared with FCN-8s and SegNet, they require less training time than DeepResUnet, but the inference time of DeepResUnet is shorter than that of FCN-8s, close to that of SegNet. From the viewpoint of accuracy improvement and reducing computing resources, such a minor time increase should be acceptable. Overall, DeepResUnet achieves a relative trade-off between the model performance and complexity.

**Table 4.** Complexity comparison of FCN-8s, SegNet, DeconvNet, U-Net, ResUNet, DeepUNet and the proposed DeepResUnet.

| Model | Parameters (m) | Training Time (Second/Epoch) | Inference Time (ms/image) |
|---|---|---|---|
| FCN-8s [28] | 134.27 | 979 | 86.1 |
| SegNet [29] | 29.46 | 1192 | 60.7 |
| DeconvNet [31] | 251.84 | 2497 | 214.3 |
| U-Net [30] | 31.03 | 718 | 47.2 |
| ResUNet [51] | 8.10 | 1229 | 55.8 |
| DeepUNet [57] | 0.62 | 505 | 41.5 |
| DeepResUnet | 2.79 | 1516 | 69.3 |

### 4.4. Applicability Analysis of DeepResUnet

To further explore the applicability of DeepResUnet, the urban area of Waimakariri, New Zealand was used to test the effectiveness of DeepResUnet. The aerial images of Waimakariri were taken during 2015 and 2016, with a spatial resolution of 0.075 m. The corresponding building outlines were also provided by the website (https://data.linz.govt.nz). The aerial image of Waimakariri is 13,526 × 12,418 pixels. Note that DeepResUnet was trained using the aerial images of Christchurch City, New Zealand,

and we did not train DeepResUnet again on other images. Additionally, any pre- and post-processing methods were not applied when testing the aerial images of Waimakariri.

The results of the Waimakariri area are presented in Figure 7. Many false negatives (green) appeared in the segmentation results of SegNet model and many misclassifications (false positive) in the segmentation results of DeconvNet model. It is obvious that FCN-8s and DeepResUnet outperformed the other approaches. Neither ResUNet nor DeepUNet performed well in this new urban area, indicating the relatively poor applicability of these combined models. As exhibited in last three rows of Figure 7, few false positives (red) and false negatives (green) were obtained in the segmentation results of DeepResUnet model. DeepResUnet accurately identified small-size and irregularly shaped buildings. Quantitative results are provided in Table 5. The overall performance of DeepResUnet in this study was the best among deep models, followed by FCN-8s and ResUNet. The performance of these deep models was basically consistent with the testing results of Christchurch City, indicating that DeepResUnet has the ability to identify urban buildings and it can be applied to dense urban areas such as big cities and even megacities.

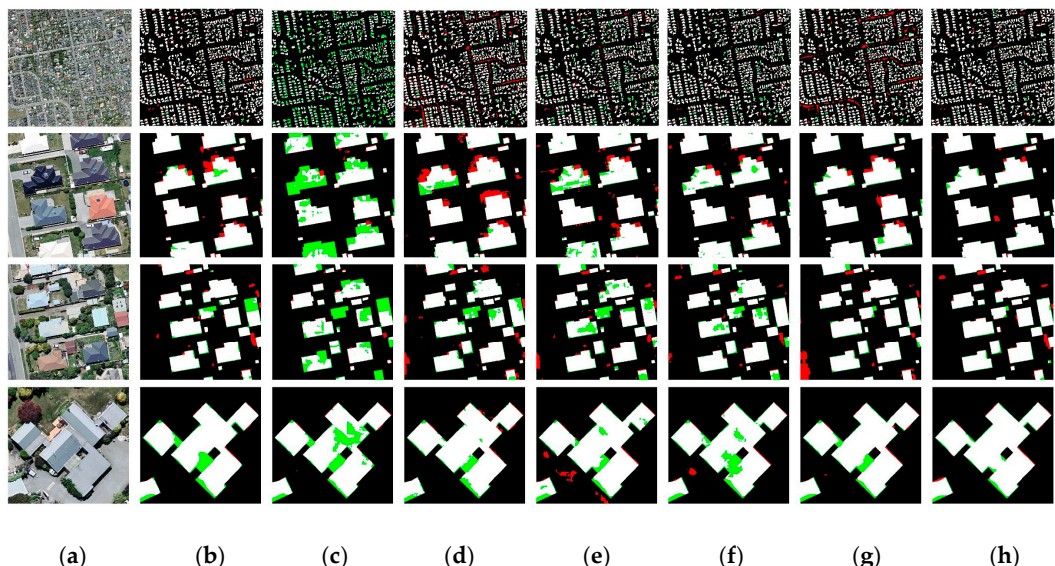

|  (a)  |  (b)  |  (c)  |  (d)  |  (e)  |  (f)  |  (g)  |  (h)  |

**Figure 7.** Visual comparison of segmentation results by seven approaches (for an urban area of Waimakariri, New Zealand). The first row shows the overall results, and the last three rows exhibit the zoomed-in results. In the colored figures, the white, red and green colors represent true positive, false positive and false negative predictions, respectively. (**a**) Image. (**b**) FCN-8s. (**c**) SegNet. (**d**) DeconvNet. (**e**) U-Net. (**f**) ResUNet. (**g**) DeepUNet. (**h**) DeepResUnet.

**Table 5.** Quantitative comparison of five conventionally used metrics (for an urban area of Waimakariri, New Zealand) obtained from the segmentation results by FCN-8s, SegNet, DeconvNet, U-Net, ResUNet, DeepUNet and the proposed DeepResUnet, where the values in bold format are the highest numbers for corresponding metrics.

| Models | Precision | Recall | F1 | Kappa | OA |
|---|---|---|---|---|---|
| FCN-8s [28] | 0.8831 | **0.9339** | 0.9078 | 0.8807 | 0.9581 |
| SegNet [29] | **0.9475** | 0.6174 | 0.7477 | 0.6944 | 0.9079 |
| DeconvNet [31] | 0.8004 | 0.9135 | 0.8532 | 0.8080 | 0.9306 |
| U-Net [30] | 0.8671 | 0.8621 | 0.8646 | 0.8263 | 0.9403 |
| ResUNet [51] | 0.9049 | 0.8895 | 0.8972 | 0.8683 | 0.9549 |
| DeepUNet [57] | 0.8305 | 0.9219 | 0.8738 | 0.8356 | 0.9412 |
| DeepResUnet | 0.9101 | 0.9280 | **0.9190** | **0.8957** | **0.9638** |

The bolded numbers indicate the largest number in the column, easier to find out which one performs better in this format.

*4.5. Limitations of Deep Learning Models in This Study*

Although deep learning models achieved impressive results in semantic segmentation, some limitations exist in those models due to the complexity of remotely sensed images. As can be seen in Figure 8, in some particular areas, many false positives appeared in the result of building segmentations of all networks used in this paper. This problem may be caused by the phenomenon of "different objects with the same spectral reflectance" or "same objects with the different spectral reflectance". Actually, this phenomenon extensively exists in remotely sensed images. For DCNN-based methods, it is difficult to learn robust and discriminative representations from insufficient training samples and to distinguish subtle spectral differences [32]. In addition, many buildings were not fully identified by deep learning models because of roadside trees or shadows. This is also a challenge for DCNN-based methods.

Recently, some studies [63–66] found that the effective fusion of color imagery with elevation (such as DSM) might be helpful to resolving these problems. The elevation data containing the height information of the ground surface make it easy to discriminate the building roofs from impervious surfaces. Additionally, only three-band (red, green, blue) images were used to extract buildings. The near-infrared band was not used in the study which might be helpful to identify vegetation. In the future work, the use of elevation data and multispectral images (including the near-infrared band) will be considered for alleviating these issues.

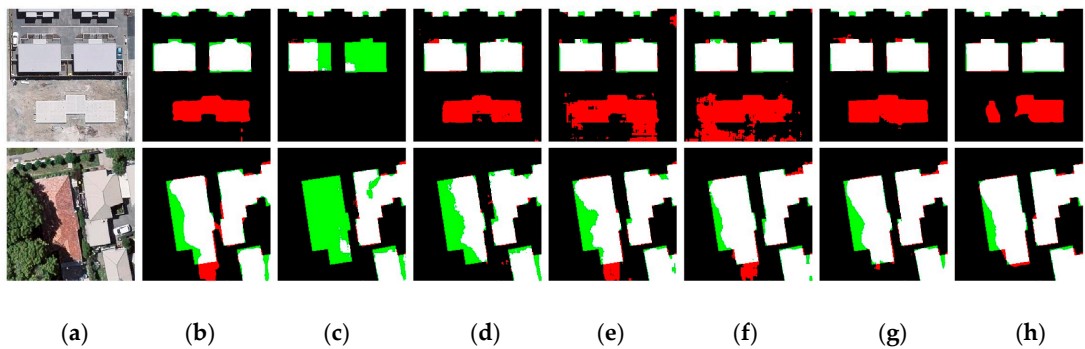

| (a) | (b) | (c) | (d) | (e) | (f) | (g) | (h) |

**Figure 8.** Visual comparison of urban building segmentation results by using different networks. In the colored figures, the white, red and green colors represent true positive, false positive and false negative predictions, respectively. (**a**) Image. (**b**) FCN-8s. (**c**) SegNet. (**d**) DeconvNet. (**e**) U-Net. (**f**) ResUNet. (**g**) DeepUNet. (**h**) DeepResUnet.

## 5. Conclusions

An end-to-end DCNN, denoted as DeepResUnet, for VHR image semantic segmentation of urban buildings at pixel scale, was proposed by adopting the architecture of U-Net as the basic structure. Specifically, the proposed DeepResUnet contains two sub-networks, that is, a cascade down-sampling network that is used to extract building feature maps from the VHR image, and an up-sampling network that is used to reconstruct the extracted feature maps of buildings back to the same size of input VHR image. To reduce gradient degradation, deep residual learning was incorporated in the proposed network.

To evaluate the performance of DeepResUnet, six existing state-of-the-art deep networks, including FCN-8s, SegNet, DeconvNet, U-Net, ResUNet and DeepUNet were selected for comparison in the exact same experiment environment, both visually and quantitively. Each network was trained from scratch rather than being pretrained before the experiments. One of the advantages of DeepResUnet was that it requires far less parameters than most methods except DeepUNet. However, it does require slightly longer inference time than some other networks. For visual comparison, it was clear that all the seven tested networks were capable of extracting the regular-shape buildings, such as rectangle and square shaped buildings. However, other networks were less capable in accurate extraction of the irregularly shaped buildings, demonstrating that the proposed DeepResUnet outperformed the

other six networks in a way that fewer false negatives and false positives appeared in the semantic segmentation image of urban buildings. Five conventionally used criteria, that is, precision, recall, F1 score (F1), Kappa coefficient and overall accuracy (OA), were used to evaluate the performance of the networks quantitatively, where DeepResUnet outperformed all the others because it suppressed false negatives, especially in semantic segmentation of irregular-shape and small-size buildings with higher accuracy and shape entirety. DeepResUnet is relatively better at suppressing false negatives as shown by its superior recall. Compared with the U-Net, DeepResUnet increased the F1 score, Kappa coefficient and overall accuracy by 3.52%, 4.67% and 1.72%, respectively. Additionally, DeepResUnet was further tested using the aerial images of an urban area of Waimakariri, New Zealand, further indicating the effectiveness and applicability of DeepResUnet.

More research is needed to improve DeepResUnet to better discriminate different objects with similar spectral characteristics or the same objects with different spectral characteristics. To some extent, the proposed network was designed for pixel-level urban building semantic segmentation and it may be applied to dense urban areas such as big cities and even megacities. As a continuation of this work, the fusion of image with elevation data (such as DSM) may be considered in the future to refine the performance of the proposed method.

**Author Contributions:** W.Z., Y.Y. and Z.Z. conceived this research. Y.Y. and Z.Z. performed the experiments, analyzed the results and wrote the paper. W.Z., C.Z., W.L. and T.Z. gave comments and modified the manuscript.

**Funding:** This research was funded by the National Key Research and Development Program of China, grant number 2016YFA0602302 and 2016YFB0502502.

**Acknowledgments:** The aerial images are provided by National Topographic Office of New Zealand (https://data.linz.govt.nz).

**Conflicts of Interest:** The authors declare no conflict of interest.

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
