# Peer review of "Semantic Segmentation of Urban Buildings from VHR Remote Sensing Imagery Using a Deep Convolutional Neural Network"

_remotesensing, doi:10.3390/rs11151774_

Round 1
Reviewer 1 Report
This paper used U-Net and Resblock for semantic segmentation of urban buildings in remote sensing image, and the proposed method achieved good segmentation results.
Some references about building extraction/segmentation should be added.
It is difficult for us to reasonably evaluate the contribution of this paper without the performance comparison between the proposed method and other combined approaches of the U-Net with the deep ResNet.
Analysis on why DeepResUnet takes longer inference time should be part of Section 4.3 Complexity comparison of deep learning models. Since the authors have analyzed this issue in the last round of responses, I think it is unreasonable to avoid this issue in this paper and regard it as the content of future research.
Reviewer 2 Report
A deep convolutional neural network is proposed to segment urban buildings captured in images. I think the proposal is interesting because the authors perform experiments on large image.
But I have some questions.
How is trained the neural networks? It is not clear to me which the size of the input images is. are the large images downsampled to 256x256 pixels?
It would be interesting the authors show the running time of the algorithm
Reviewer 3 Report
C1: Abstract and conclusions must be extended with obtained precision.
C2: [see rows 103-104, 227-228, 436-441] Firstly, description of the raw data format is not clear. Is it spectral images (reflectance of surface [0..1]) or orthoimages (RGB image [0..255])? Secondly, the mention about orthoimage generation is not clear too. Is it completed by authors or by data provider? And it would be useful to mention the method of orthoimage generation from aerophoto, if this subject is discussed.
C3: the resolution of image is 0.075 m, the input of DCNN is 256x256. Therefore, the input region is 19.2 m. Considering the application, it is small region comparable with sizes of buildings. Additionally, Fig. 8 provides false detected building. It would be great to provide some comments in discussion about developed DCNN application for real tasks (open challenges and possible solutions).
C4: it is considered "bad style" to use sentences with {I, we, author, authors} in scientific publications. Good style is neutral form usage (without them).
Round 2
Reviewer 1 Report
All my comments and suggestions have been properly addressed, but there are still a few grammar or spelling mistakes. The authors need to check the manuscript again and again.
This manuscript is a resubmission of an earlier submission. The following is a list of the peer review reports and author responses from that submission.
Round 1
Reviewer 1 Report
The presented work addresses a topic of great interest to the scientific community as is the application of neural network systems in order to obtain information from a set of aerial images (in this case, from orthoimages and oriented to the automatic extraction of buildings ). Undoubtedly, this is a topic of maximum interest for the automation of the cartographic updating processes and the detection of changes in these urban areas.
The work presents an adequate structure, in which an adequate justification of the interest of the work is made and a good methodological introduction (although it is clear to have in mind that it is not a simple subject to be addressed), on the other hand, it presents an example practical application of the methodology and a complete analysis of results, comparatively to other methods. It also presents an adequate list of bibliographical references that allows the reader to delve more deeply into some of the aspects raised in the work.For my part, I consider that the work has been carried out with a high scientific and technical rigor, and the presentation of its results and conclusions are correctly supported in its development.
In this sense, I consider that the work is susceptible to be published in Remote Sensing in the current format.
Reviewer 2 Report
This paper proposes a novel approach based on deep learning and semantic segmentation applied to VHR imagery of urban areas. It is a good contribution. Adequate novelty to a given problem, good and well-balanced experimental design, convincing performances clearly outperforming alternatives methods with several metrics. In addition, the manuscript is well written and fairly illustrated. But there are some points that require improvement, before this manuscript can be accepted for publishing
- The motivation for proposing this novel approach is a bit unclear. Although it can be derived from the review of the state-of-the-art and along the reading of the text, there is no reason to not present it explicitly. Improve this issue in the introduction section.
- It seems that many of the TP and FN errors (seen in Figure 7) occur close to the buildings, although less evident in DeepResUnet than in the alternative methods. Are they related incorrect (or dubious) segmentation? And are they related to some specific elements as shadows, vegetation, stairs or any other? It would be interesting to increase the level of detail in the related discussion.
- In addition, the limitations indicated (section 4.4) are not exactly of the deep learning methods ‘per se’ but rather to the spectral data used (only RGB). I would expect to see noticeable improvements of the same deep learning methods only by using a multispectral image (maybe one single additional NIR band would be enough). The use of enlarged spectral data, together with ancillary data as the elevation (like mentioned) may overcome some of the problems faced by these methods. So, I would rewrite a bit this subsection, changing also the title to something more specific like ‘Limitations of deep learning models in this study’.
- Also, the semantic of the writing (not of the method…) should be improved in this subsection. For instance, ‘objects with the same spectrum’, and similar descriptions’ should be described as ‘objects with the same spectral reflectance’. The use of ‘impervious surfaces’ is also not very clear, are you talking only about man-made structures, or this also includes natural exposed areas, for instance, bare soil)? Clarify it.
- Although the manuscript is globally well written, make a careful review to improve some specific points and to correct mistakes and typos. Some of them are indicated below as example, I’m not being exhaustive:
18: ‘semantic’ instead of ‘sematic’
20. ‘proposes’ instead of ‘proposed’
154. ‘It is worthwhile to mention…’ should be ‘It is worthwhile mentioning…’
158. ‘As Figure 1 exhibited…’ should be ‘As Figure 1 exhibits…’ or ‘As exhibited in Figure 1 …’
179. ‘As Figure 1 illustrated…’ should be ‘As Figure 1 illustrates …’ or ‘As illustrated in Figure 1 …’
Reviewer 3 Report
An improved U-net is presented in this manuscript by embedded the Unet with residual learning blocks. The improved U-net show better performance than four state-of-the-art methods in terms of semantic segmentation of a VHR remote sensing image with size of 1600*2400. In other words, the improved U-net is a general neural network, which is not specific to urban building classification.
Although the improved U-net has been described clearly in this manuscript, the authors did not clearly describe the major contribution in terms of semantic segmentation of URBAN buildings.
On the one hand, both U-net and residual learning blocks have been very familiar to the deep learning society. What is new in your method?
On the other hand, the authors did not validate the improved method in terms of solving the real challenge of urban buildings classification. In other words, what is the real problem that was solved in this manuscript?
In the abstract, the author claimed that the DeepResUnet outperform the other methods especially in labeling irregular-shape and small-size buildings. However, the authors did not show specific evidences in terms of statistic results.
Reviewer 4 Report
This paper used U-Net and Resblock for semantic segmentation of urban buildings in remote sensing image, and the proposed method achieved good segmentation results. However, the novelty is not obvious and experiment is insufficient. In this context, this paper must be improved before publication in Remote Sensing for the following reasons:
The motivation that why Res U-Net is adopted for VHR remote sensing image semantic segmentation is not clear. The structure based on U-Net and Resblock is similar to the models proposed in reference [45,51-52]. Comparisons need to be added.
Only one area is used for experiment, and most part of the image are buildings. More experiment results should be presented to demonstrate the performance of the proposed method.
Problems and difficulties on urban buildings extraction of VHR images need to be expanded. Contents on road and land cover segmentation is not necessary and could be deleted.
Some statements of this article need to be explained or add some references. For example, in section 2.2, “Adding a convolutional layer can alleviate the problem of spatial information lose caused by pooling operation (why?)” and “Max-pooling layer can keep scaling invariant”.
In table 4, parameters of DeepResUnet are much less than U-Net, but training and inference take more time than U-Net. The authors need to explain the reason.